# Neuroprotective Effects of Quercetin in Alzheimer’s Disease

**DOI:** 10.3390/biom10010059

**Published:** 2019-12-30

**Authors:** Haroon Khan, Hammad Ullah, Michael Aschner, Wai San Cheang, Esra Küpeli Akkol

**Affiliations:** 1Department of Pharmacy, Abdul Wali Khan University, Mardan 23200, Pakistan; hamm.swabian@gmail.com; 2Department of Molecular Pharmacology, Albert Einstein College of Medicine, Forchheimer 209, 1300 Morris Park Avenue, Bronx, NY 10461, USA; michael.aschner@einstein.yu.edu; 3Institute of Chinese Medical Sciences, State Key Laboratory of Quality Research in Chinese Medicine, University of Macau, Macau, China; annacheang@um.edu.mo; 4Department of Pharmacognosy, Faculty of Pharmacy Gazi University, 06330 Etiler/Ankara, Turkey; esrak@gazi.edu.tr

**Keywords:** quercetin, polyphenols, Alzheimer’s disease, mechanistic insights, clinical directions

## Abstract

Quercetin is a flavonoid with notable pharmacological effects and promising therapeutic potential. It is widely distributed among plants and found commonly in daily diets predominantly in fruits and vegetables. Neuroprotection by quercetin has been reported in several in vitro studies. It has been shown to protect neurons from oxidative damage while reducing lipid peroxidation. In addition to its antioxidant properties, it inhibits the fibril formation of amyloid-β proteins, counteracting cell lyses and inflammatory cascade pathways. In this review, we provide a synopsis of the recent literature exploring the relationship between quercetin and cognitive performance in Alzheimer’s disease and its potential as a lead compound in clinical applications.

## 1. Introduction

Alzheimer’s disease (AD) contributes to 60–80% of total dementia cases, and it mostly affects elder people (65 years of age or older) [1]. The pathogenesis of AD is typically associated with the accumulation of amyloid-β (Aβ) aggregates and the hyperphosphorylation of tau proteins, leading to neurofibrillary tangles (NFTs) and synaptic dysfunction [2,3,4]. Around 35.6 million people worldwide are estimated to be affected with AD, with a prevalence rate of 4.6 million new cases each year. The prevalence rate of AD increases with age: the rate doubles every 5 years from 60 years of age [5,6].

Early studies led to the cholinergic deficit hypothesis of AD, which states that deficiency in acetylcholine is the main cause of the disease. In the pursuit of drugs that are able to restore acetylcholine levels, the first acetyl cholinesterase inhibitors were developed, including tacrine. Since then, other drugs in the same class have been pursued, namely donepezil, rivastigmine, and galantamine. Current AD therapy consists of cholinesterase inhibitors and *N*-methyl-d-aspartate (NMDA) antagonists, including memantine. Acetyl cholinesterase inhibitors prevent the hydrolysis of acetylcholine, and memantine modulates NMDA receptor activity, causing a reduction in excitatory glutamate signals. However, these drugs offer little palliative effects, and they also have numerous undesirable safety profiles with a number of adverse side effects [7,8]. Acetyl cholinesterase inhibitors are associated with gastrointestinal side effects such as nausea, diarrhea, and abdominal pain, as well as urinary incontinence, insomnia, and nightmares. The use of tacrine has been limited because of its poor bioavailability and reported hepatotoxicity. Memantine is clinically less effective compared to acetyl cholinesterase inhibitors. Additionally, these drugs are not targeting the root cause of the disorder [9].

Phytochemicals are best known to reduce the risk of chronic diseases, such as cardiovascular diseases, hypertension, diabetes, and cancers [10,11,12]. Flavonoids are the most diverse group of phytochemicals and are widely distributed in higher plants with outstanding therapeutic potential [13,14,15,16]. Flavonoids are further divided into six classes on the basis of their chemical skeleton: flavanols, flavanones, flavones, flavonols, isoflavonoids, and anthocyanidins [17]. While targeting multiple targets, they have been proven beneficial in the prevention of neurodegenerative disorders and may delay the process of neurodegeneration [18]. Flavonoids are extensively studied for their antioxidant and anti-inflammatory activities, both of which are important in triggering the pathogenesis of AD [19]. Studies suggested that flavonoids are capable of crossing the blood–brain barrier (BBB), which makes them potential agents in preventing neurodegenerative disorders; however, different flavonoid subclasses differ in their ability to cross the BBB [20,21]. In the case of AD, their efficacy is attributed to the reduction of Aβ toxicity and decreasing oxidative stress [22,23]. Nevertheless, anti-AD effects of certain flavonoids, such as myricetin, rutin, fisetin, catechins, quercetin, kaempferol, and apigenin have been reported [24,25,26,27].

Quercetin is one of the most potent antioxidants of plant origin and is one of the predominant flavonoids found more commonly in edible plants [28]. It belongs to the flavonols class of flavonoids, representing a major class of polyphenols. The dietary intake of total flavonoids is estimated to be 200–350 mg/day, and the intake of quercetin is 10–16 mg/day. The recommended dosage of quercetin aglycone as a dietary supplement is 1 g/day [29,30,31]. It exhibits numerous beneficial effects on human health, acting as anti-carcinogenic, anti-inflammatory, anti-infective, and psychostimulant agent. It also inhibits lipid peroxidation and platelet aggregation, and it stimulates mitochondrial biogenesis [32]. Several studies have reported on the neuroprotective effects of quercetin, both in vitro and in vivo models of neurodegenerative disorders, such as cognitive impairment [33], ischemia, traumatic injury [34], Parkinson’s disease (PD) [35], and Huntington’s disease (HD) [36]. The aim of the present review is to provide a summary of the recent literature exploring the relationship between quercetin and cognitive performance. Our primary focus is on the chemical basis and pharmacology of quercetin and its anti-AD mechanisms.

## 2. Chemistry of Quercetin

A flavonoid is structurally diphenyl propane containing 15 carbon atoms in its structure (Figure 1A). It contains a close heterocyclic pyran ring in addition to two benzene rings. The term 4-oxo-flavonoid (Figure 1B) is often used to describe flavonoid containing a carbonyl group on position C-4 of ring C. Substitution and oxidation in the heterocyclic pyran ring classifies flavonoids into subclasses, namely flavones, flavonols, flavanones, flavononol, isoflavones, and flavan-3-ols. Substitution in the benzene rings of flavonoid structures leads to differences in individual compounds within specific classes [37,38]. Quercetin belongs to the flavone class of flavonoids having a chemical formula of C_15_H_10_O_7_. Its IUPAC name is 3,3′,4′,5,7-pentahydroxyflavanone or 3,3′,4′,5,7-pentahydroxy-2-phenylchromen-4-one (Figure 2). Quercetin contains an OH group at positions 3, 5, 7, 3′, and 4′. Quercetin is an aglycone, (lacking an attached sugar molecule). Attaching a glycosyl group (glucose, rhamnose, or rutinose) most commonly at position 3 replacing the OH group leads to the formation of quercetin glycoside. Quercetin is insoluble or sparingly soluble in water, while it is quite soluble in alcohol and lipids. A glycosyl group increases its water solubility, and thus the quercetin glycoside is soluble in water [32,39]. In addition to antioxidant activities, multiple OH groups in the structure of quercetin may also lead to its photodegradation. Dall’Acqua et al. (2012) reported that OH groups at positions 3, 3′, and 4′ are mainly involve in photolability, while OH groups at positions 5 and 7 do not play a crucial role in the photo-oxidative mechanism [40].

Several flavonoids have anti-inflammatory activity, among which flavanones such as naringenin have weak activity, while flavonols including quercetin, kaempferol, and myricetin have strong anti-inflammatory activity. Flavonols are more active against phospholipase A-2, given the presence of the C-ring-2,3 double bond. Furthermore, the glycosylation of quercetin, such as rutin, reduces the anti-inflammatory activity of quercetin [41]. Glycosylation at position 3 of quercetin leads to a decrease in the free radical neutralizing activity. Glycosylation also decreases the acetyl cholinesterase inhibition activity of quercetin [42]. The methylation of quercetin at positions 4′ and 7 results in improved anticancer activities. The replacement of OH groups with O-methylated moieties also increases the metabolic stability of a compound [43]. The presence of a double bond between C2 and C3 as well as the OH group at ring B is essential for the thrombin inhibition activity of quercetin. Substituting the OH group with an OCH_3_ group in ring B and ring C reduces the thrombin inhibition activity, whereas replacement of the OH groups in ring A with an OCH_3_ group improves the thrombin inhibition activity of quercetin [44].

Quercetin may act as a pro-oxidant phytochemical by generating reactive oxygen species (ROS) and reactive electrophilic quinine type metabolites because of the presence of catechol moiety in the B ring, C-ring-2, 3 double bond, and free OH group at the C-3 position [45]. Furthermore, in vitro studies have confirmed that the esters-based precursors of quercetin increase the bioavailability of quercetin [46]. In considering the general structure of flavonoids, OH group substitution at positions C5, C7, and C3, the substitution of OCH_3_ or H at C3′, and OH or OCH_3_ substitution at position C4′ improve their neuroprotective activities as confirmed by in vitro studies, utilizing neuronal cell cultures [47].

## 3. Sources

The name quercetin has been used since 1857. It is derived from Latin *Quercetum,* after *Quercus* (oak) [48]. It is a flavonoid that is most commonly occurring in higher plants and in the glycosidal form, such as rutin (quercetin-3-*O*-rutinoside), isoquercetin (quercetin-3-*O*-glucoside), and hyperin (quercetin-3-*O*-galactoside). It can also be isolated in free-form from leaf surfaces, fruits, or bud extracts. Plant families rich in quercetin are Compositae, Passiflorae, Rhamnaceae, and Solanaceae [49]. Onions, asparagus, red leaf lettuce, apples, capers, and berries contain relatively high concentrations of quercetin [29,50].The botanical sources of quercetin have been summarized in Table 1.

## 4. Pharmacokinetic Parameters of Quercetin

Studies have revealed that the low bioavailability of quercetin limits its use for therapeutic purposes, although it has a wide range of pharmacological properties. Low solubility, poor absorption, and rapid metabolism are major aspects of the low bioavailability of quercetin [80]. Ader et al. (2000) have explored the bioavailability and metabolism of quercetin in pigs [81]. Quercetin was administered to each animal in a single intravenous dose of 0.4 mg/kg body weight and a single oral dose of 50 mg/kg body weight one week later. Blood samples were collected to analyze the pharmacokinetic parameters of quercetin. The apparent bioavailability was 0.54 ± 0.19% when considering free quercetin alone, 8.6 ± 3.8% including conjugated quercetin, and 17.0 ± 7.1% including quercetin metabolites (kaempferol, tamarixetin, and isorhamnetin). The authors also reported that the conjugation of quercetin to glucuronic acid and sulfuric acid preferentially occurs in the intestinal wall.

Moon et al. (2008) have studied the pharmacokinetics of quercetin in human beings [82]. Quercetin was administered to subjects in doses of 500 mg three times daily, and the plasma and urine samples were collected from subjects to analyze the concentration of quercetin aglycone and metabolites. The average peak plasma concentration reported after the administration of quercetin at 500 mg three times daily was 463 ng/mL at 3.5 h. Re-entry peaks on plasma concentration versus time curves showed enterohepatic recirculation. The oral clearance of quercetin was reported to be high (3.5 × 10^4^ L/h) with an average half life of 3.5 h. The urinary recovery of quercetin aglycone and conjugated metabolites were 0.05% to 3.6% and 0.08% to 2.6%, respectively. Previously, Day et al. (2001) while studying the identification of quercetin metabolites in plasma have reported that quercetin is found exclusively as a glucuronated or sulfate conjugate in plasma after oral administration [83].

Moreover, quercetin aglycone is less effective to cross the BBB, as it is a substrate for efflux transporter p-glycoprotein. On the other hand, quercetin conjugates are also predicted to less effectively cross the BBB because of high polarity [84,85]. However, the literature supported that some glucuronides are effectively crossing the BBB because of aided active transport [86]. Kawei et al. (2008) revealed that at sites of inflammation, the intracellular deconjugation of quercetin 3-*O*-glucuronide by β-glucuronidase increases, and this phenomena is more relevant inside the central nervous system (CNS) in the case of neurodegenerative disorders, rather than being of relevance in the penetration of the BBB [87].

## 5. Pathophysiological Mechanisms of Alzheimer’s Disease

AD is mainly characterized by the accumulation of extracellular amyloid plaques, which are known as senile plaques. The amyloid plaques are associated with neuroinflammatory changes as well as neurofibrillary tangles, affecting processes that are essential in maintaining neuronal health: communication, metabolism, and repair. Aβ is produced by the sequential cleavage of amyloid β precursor protein by β and γ secretases [88,89]. The amyloid precursor protein (APP) exists in three different forms: the shortest isoform, APP695 mostly expressed in neurons, whereas two other isoforms APP751 and APP770 are expressed in glial cells, such as astrocytes. The physiological role of APP is to stimulate cell growth and proliferation. Aβ plaques lead to neuronal loss in several brain regions, including the entorhinal cortex, hippocampus, neocortex, amygdale, and subcortical areas [90,91]. Two different enzymatic processes of APP occur in the cells: the non-amyloidogenic and the amyloidogenic pathways. In the non-amyloidogenic processing pathway, the APP is cleaved by α and γ-secretases, which yields long secreted forms of APP (sAPPα) and C-terminal fragments. In the amyloidogenic processing pathway, APP is cleaved by β and γ-secretases, which yields long secreted forms of APP (sAPPβ), C-terminal fragments, and Aβs. The Aβ fragment is chemically sticky-forming plaques, and blocks cell-to-cell signaling via synapses, in turn leading to neuronal cell death [92,93].

Neurofibrillary tangles (NFTs) are produced by the hyperphosphorylation of microtubule-associated proteins, which are known as tau [94], representing another key feature of AD. The excessive phosphorylation of tau proteins decreases their binding ability to microtubules, resulting in the formation of NFTs [95]. Both the density and neuroanatomic localization of NFTs are important determinants in the pathogenesis of AD. NFTs in neocortical regions are commonly associated with severe cognition impairment [96]. Aβ induces the phosphorylation of tau proteins, thus triggering the formation of NFTs and causing neuronal cell death by increasing oxidative stress, altering calcium homeostasis and excitotoxicity [90]. Imbalance between Aβ production and clearance leads to the formation of Aβ aggregates, especially in late onset AD. The impairment of Aβ clearance most commonly occurs in late onset AD. Heparan sulfate proteoglycans (HSPGs) are most critical in the pathogenesis of AD by affecting Aβ metabolism and decreasing Aβ clearance. HSPGs bind to Aβ and accelerate its aggregation, mediating the neurotoxicity and neuroinflammation induced by Aβ. Under normal conditions, HSPGs perform various important physiological functions in development, growth factor signaling, cell proliferation, adhesion, migration, and homeostasis [97].

Oxidative stress commonly occurs in AD, contributing to neurodegeneration. ROS generation triggers Aβ aggregation. Oxidative stress is inherent to mild cognitive impairment (MCI) in the progression of AD. Patients with MCI usually develop cognitive impairment with a minimal impairment of instrumental activities of daily life, and it can be the first cognitive expression of AD. Alteration in the phosphorylation of proteins, such as heme oxygenase-1 and biliverdin reductase A, has been noted to occur, affecting the signaling of the most critical antioxidant pathways [98,99,100]. This leads to mitochondrial damage, which is concomitant with the decreased activity of mitochondrial energy-related proteins, including pyruvate dehydrogenase complex and alpha-ketoglutarate dehydrogenase. Defective mitochondria, in turn, trigger the generation of high levels of reactive oxygen species (ROS), for which antioxidant defenses may be deficient [101,102,103]. As such, the process proceeds unabated, as in a brush fire.

Neuroinflammation is a hallmark in the pathogenesis of AD. The innate immune cells of the brain are fast to respond to systemic events, mostly in aged and diseased brains. Misfolded and aggregated proteins binding to pattern recognition receptors on microglia and astroglia trigger an immune response, releasing a number of inflammatory cytokines and chemokines [104,105]. In contrast to oxidative stress, which is short lived, chronic inflammation is long lasting, resulting in the sustained release of inflammatory mediators.

Microglia surrounding senile plaques is commonly activated in AD, resulting in the upregulation of human leukocyte antigen-DR (HLA-DR). In addition, the generation of inflammatory mediators may lead to microglial activation and neurotoxicity secondary to the CD14-dependent process. Despite having phagocytotic activity, microglia fail to phagocytose Aβ due to the presence of inflammatory cytokines and various extracellular matrix proteins [106]. Receptor of advanced glycation end products (RAGE) activation leads to neurodegeneration as it triggers an increase in inflammatory mediators and oxidative stress. RAGE activation also leads to activation of downstream regulatory pathways such as the NF-kB, STAT, and JNK pathways [107].

Acetylcholinergic, glutamatergic, and serotonergic neurons are mostly affected in AD. Early histological changes involve the loss of cholinergic neurotransmission in the cerebral cortex. The latter, along with the hippocampus, receive the highest cholinergic input, and the loss of cholinergic neurons in these regions results in cognitive deficits and memory impairment. Choline acetyltransferase activity is greatly reduced in individuals with AD. It has also been shown that acetyl cholinesterase (AChE) accelerates the aggregation of Aβ [108]. The degeneration of serotonergic neurons in the raphe nucleus and noradrenergic neurons in the locus coeruleus mediates on cognitive symptoms associated with AD [109]. Figure 3 illustrates the pathogenesis of Alzheimer’s disease.

## 6. Neuroprotective Efficacy of Quercetin

The neuroprotective effects of quercetin have been extensively studied. At low micromolar concentrations, it antagonizes cell toxicity by oxidative stress in neurons. It suppresses neuroinflammatory processes by downregulating pro-inflammatory cytokines, such as NF-kB and iNOS, while stimulating neuronal regeneration. After absorption, quercetin metabolites are glucuronated, methylated, or sulfated, and all have been shown to afford neuroprotection. The neuroprotective efficacy of quercetin has been studied in both in vitro and in vivo models [50,110,111]. A study using drug screening in *Caenorhabditis elegans* nematodes with neuronal expression of human exon-1 huntingtin (128Q) and mutant Htt striatal cells derived from knock-in HD mice, concluded that isoquercetin improved motor functions in acute spinal cord injury, reduced α-synuclein fibrillization, reduced hippocampal neuronal cell death, improved synaptic plasticity, and reversed histopathological hallmarks of AD [112]. Quercetin also protects against mitochondrial dysfunction and progressive dopaminergic neurodegeneration by activating PKD1-Akt cell survival signaling axis Cell Culture and MitoPark transgenic mouse models of Parkinson’s disease [113].

Quercetin has shown therapeutic efficacy, improving learning, memory, and cognitive functions in AD [114]. Khan et al. (2009) and Shimmyo et al. (2008) concluded that quercetin administration resulted in the inhibition of AChE and secretase enzymes using in vitro models, thus preventing the degradation of acetylcholine, and decreasing Aβ production, respectively [115,116]. Sabogal-Guáqueta et al. (2015) have been reported that quercetin administration reverses extracellular β-amyloidosis and decreases tauopathies, astrogliosis, and microgliosis in the hippocampus and amygdale, thus protecting cognitive and emotional function in age triple transgenic Alzheimer’s disease model mice [117]. Wand et al. (2014) studied the effects of the long-term administration of quercetin on cognition and mitochondrial dysfunction in a mouse model of Alzheimer’s disease. They noted that quercetin ameliorates mitochondrial dysfunction by restoring mitochondrial membrane potential, decreases ROS production, and restores ATP synthesis. It also increased the expression of AMP-activated protein kinase (AMPK), which is a key cell regulator of energy metabolism. Activated AMPK can decrease ROS generation by inhibiting NADPH oxidase activity or by increasing the antioxidant activity of enzymes such as superoxide dismutase-2 and uncoupling protein-2. The activation of AMPK also decreased Aβ deposition, regulating APP processing and promoting Aβ clearance. These mechanisms likely account for some of the therapeutic efficacy of quercetin on cognition and the attenuation of Aβ-induced neurotoxicity [118]. Quercetin and rutin have also been reported to function as memory enhancers in scopolamine-induced memory impairment in zebrafish, thus possibly enhancing cholinergic neurotransmission [119].

## 7. Anti-Alzheimer’s Disease Mechanisms of Quercetin

### 7.1. Inhibition of AβAggregation and Tau Phosphorylation

The aggregation of Aβ is a key hallmark of AD [120]. Quercetin interferes with the formation of neurotoxic oligomeric Aβ species and displays fibril destabilizing effects on preformed fibrillar Aβ, reversing Aβ-induced neurotoxicity [110]. The structure of efficient polyphenolic inhibitors of Aβ contains two aromatic rings with two to six atom linkers. The aromatic rings contain a minimum number of three hydroxyl groups, which play an important role in fibril inhibition through hydrophobic interaction between the aromatic rings with β-sheet structures, forming hydrogen bonds. The phenolic hydroxyls increase the electron density in the aromatic rings, which may increase the binding of quercetin with the aromatic amino acids of the peptide beta-sheet structures. Quercetin possesses these structural requirements containing hydrophobic moieties and thus arrests fibril formation. The more hydroxyl groups present in the structure of the molecule, the higher its anti-amyloidogenic activity [121,122]. It is also suggested that the catechol structure may be auto-oxidized to form o-quinone on ring B, which then forms an O-quinone-Aβ42 adduct by targeting Lys residues at positions 16 and 28 of Aβ42. This phenomenon explains why quercetin has higher Aβ aggregation inhibitory activities compared to kaempferol, morin, and datiscetin [123].

Quercetin is also reported from in vitro and in silico studies to inhibit beta-secretase-1 (BACE-1) enzyme activity through the formation of hydrogen bonds. The OH group at position C-3 has a significant role in BACE-1 inhibition [116]. It has been documented from in vitro and molecular docking studies performed by Paris et al. (2011) that NF-kB regulates the production of Aβ by regulation of the β cleavage of APP, and that the quercetin-induced inhibition of NF-kB affects the regulation of BACE-1 expression [124]. Tauopathy commonly begins in the hippocampus, affecting hippocampal-dependent cognitive tasks followed by progression to other brain areas. Quercetin has been shown to decrease the phosphorylation of tau proteins and to inhibit the formation of NFTs in age triple transgenic Alzheimer’s disease model mice [117]. Kinases and protein phosphatases such as protein phosphatase 2A (PP2A) play a role in regulating the hyperphosphorylation of tau proteins. The hyperphosphorylation of tau proteins is mostly due to the imbalance between phosphorylation and dephosphorylation mechanisms. Thus, the inhibition of PP2A may lead to the hyperphosphorylation of tau proteins. Quercetin reverses the hyperphosphorylation of tau proteins via MAPKs and PI3K/Akt/GSK3β signaling pathways in HT22 cells (a cell line from mouse hippocampal neurons) [125].

### 7.2. Acetylcholinesterase Inhibition

The inhibition of AChE is one of the therapies most commonly pursued in the treatment of mild to moderate AD. AChE is an enzyme that is responsible for the degradation of acetylcholine, and its inhibition results in increased acetylcholine levels, thus improving the cognitive symptoms of AD. In vitro studies have shown that quercetin is a competitive inhibitor of AChE and butyrylcholinesterase (BChE). It inhibits both enzymes in a concentration-dependent manner [126]. Quercetin inhibits AChE secondary to hydrophobic interactions and strong hydrogen bonding with the enzyme, reducing the hydrolysis of ACh, thus increasing ACh levels in the synaptic cleft, as reported by Abdalla et al. (2013) using cadmium-exposed rats as a model of study [127]. Studies have linked the presence/absence of OH groups on the phenyl rings of the test compound to the inhibition of AChE and BChE, given that the OH groups form hydrogen bonds with amino acid residues at the active site of the enzyme. This phenomenon may explain the inhibitory efficacy of quercetin for both of these enzymes. Moreover, quercetin exhibits greater potency for the inhibition of AChE and BChE than its glycosidal form, rutin. Previous studies have posited the presence of a sugar moiety in the molecule is essential for its enzymatic inhibition [115,128,129]. Jung et al. (2007) studied the acetylcholinesterase inhibiting the potential of flavonoids, including quercetin isolated from *Agrimonia pilosa* and they reported the IC50 value of quercetin to be 19.8 [130].

### 7.3. Attenuation of Oxidative Stress

Oxidative stress is caused by the accumulation of ROS in cells, and is an important factor in various neurodegenerative disorders. It is the most common mechanism of age-related degenerative processes. Mitochondria are the primary site for ROS production, and mitochondrial dysfunction leads to the overproduction of ROS followed by ATP depletion and ultimately cell death [50,131]. The major source of ROS is the superoxide anion radicals generated by the electron transport chain during oxidative phosphorylation. Superoxide dismutase (SOD) converts superoxide ions to hydrogen peroxide, which can be detoxified by catalase and glutathione peroxidases [132]. It has been reported that Aβ triggers oxidative stress, leading to lipid peroxidation and protein oxidation, which results in damaged mitochondria and the dysfunction of key enzymes associated with various pathways, including glucose metabolism [133].

Among the phytochemicals, quercetin is a potent antioxidant. It has been shown to effectively reduce the concentrations of superoxide anion free radicals, and its antioxidant potential makes it a versatile choice in the management of various disorders, including AD [49]. Previous studies have shown that quercetin has direct radical scavenging action. The presence of two pharmacophores in its structure is responsible for its antioxidant activities: one is a catechol group in the B ring, and the other is the OH group at position C-3. Quercetin also modulates the cell’s own antioxidant pathways, by inducing Nrf-2-ARE and paraoxonase 2 (PON2), which is an antioxidant enzyme. Nuclear factor (erythroid-derived 2)-like 2 (Nrf-2) is an important regulator of cellular defenses against oxidative stress. Heme oxygenase-1, glutamate cysteine ligase, glutathione S-transferase, glutathione peroxidase, SOD, catalase, sulfiredoxin, and thioredoxin are enzymes downstream of Nrf-2-ARE. Thus, the activation of the Nrf-2-ARE pathway likely modulates the formation and degradation of misfolded protein aggregates in AD [50,64,134].

### 7.4. Attenuation of Neuroinflammation by Quercetin

Reducing the neuroinflammatory events in microglia might afford a beneficial strategy for the prevention of the progression of inflammatory-mediated neurodegenerative disorders. Quercetin has been reported to have anti-inflammatory actions, and is a suitable candidate among phytochemicals for future studies on its efficacy to reverse neuroinflammation [135,136]. Quercetin has already been shown to inhibit neuroinflammation by reducing nitric oxide production, iNOS gene expression in microglia, the production of inflammatory cytokines such as tumor necrosis factor-α (TNF-α), IFN-γ, interleukin-1β (IL-1β), IL-6, IL-12, and COX-2 in activated macrophages, as well as a reduction in cytokine expression as reported from in vivo studies. Quercetin also downregulates JNK/Jun phosphorylation and inhibits TNF-α production in mice, thus protecting neurons against LPS-induced inflammation [137]. Figure 4 and Figure 5 have been summarizing anti-Alzheimer’s targets and mechanistic insights of quercetin, respectively.

## 8. Anti-Alzheimer’s Disease Potential: In Vitro Studies on the Efficacy of Quercetin

Quercetin protects neurons from oxidative damage, while reducing lipid peroxidation. Furthermore, given its antioxidant properties, quercetin inhibits the fibril formation of Aβ proteins, counteracting cell lyses and inflammatory cascade pathways [7,138,139]. In vitro studies on quercetin have shown that its antioxidant activities are concentration-dependent. It acts as an antioxidant at lower concentrations (neuronal cells treated with 5 µM and 10 µM), but it possesses toxic effects at higher doses (neuronal cells treated with 20 µM and 40 µM) [140]. It also inhibits iNOS and regulates the expression of COX-2 in various models, reflecting upon its anti-inflammatory activities. Most of the absorbed quercetin is present as metabolites, including glucuronidated, methylated, and sulfated metabolites. All exert neuroprotective effects, but limited testing of these metabolites has been carried out to date [117]. It has also been reported that quercetin decreases APP maturation, thus altering Aβ synthesis and aggregation [141]. The neuroprotective effects of quercetin glycosides have also been reported, which are characterized by antagonizing changes in gene expression, such as Park2, Park5, Park7, Casp3, and Casp7 genes [142].

## 9. Anti-Alzheimer’s Disease Potential: In Vivo Studies on the Efficacy of Quercetin

The protective effects of a diet rich in polyphenols have been reported in several pathologic conditions, such as cardiovascular diseases, metabolic disorders, infections, cancers, and neurodegenerative disorders. It has been reported in several studies that quercetin exerts neuroprotective effects when administered in vivo. It protects neuronal cells from oxidative stress induced by various chemicals and prevents hippocampal apoptosis [50,143]. Quercetin improves memory, learning, and cognitive functions, and all these effects have been shown to be associated with its antioxidant properties [144]. In vivo studies using mice as an animal model have supported that quercetin increases spatial memory tasks, and decreases β-amyloidosis, tauopathies, astrogliosis, and microgliosis by increasing AMPK activity and decreasing mitochondrial dysfunction [117,118].

Keddy et al. (2012) investigated the neuroprotective and anti-inflammatory effects of the flavonoids-enriched fraction containing quercetin and its glucosides in a mouse model of hypoxic-ischemic brain injury. The study had concluded that the repeated administration of the flavonoid-enriched fraction prior to an experimental stroke produced by hypoxic-ischemia prevents the neuronal loss in the striatum and dorsal hippocampus. Due to the low bioavailability of quercetin and its glucosides, it required injection through the intraperitonial or intravenous route to achieve neuroprotective effects [145]. Tota (2010) et al. studied the effects of quercetin on cerebral blood flow and memory impairment in mice and linked the ability of quercetin to increase cerebral blood flow and energy metabolism to its memory-enhancing effects [146]. After oral administration in humans, quercetin is extensively metabolized during its absorption from the gut, affecting its bioavailability. Clinical efficacy trials/studies of quercetin have yet to be carried out, which is likely given its low BBB penetrability [147]. Furthermore, the metabolites of quercetin have long half-lives in vivo, and repeated dosing may lead to plasma accumulation [145]. These are important factors, which will need to be considered in the design of quercetin analogs for clinical studies.

## 10. Conclusions

Quercetin is a flavonoid with notable pharmacological effects and promising therapeutic potential. It is widely distributed among plants and found commonly in our daily diet, such as in fruits and vegetables. It has beneficial properties against general mechanisms of AD etiology in a variety of in vitro and in vivo models. It protects neuronal cells by attenuating oxidative stress and neuroinflammation. The anti-Alzheimer’s disease properties of quercetin include the inhibition of Aβ aggregation and tau phosphorylation. It restores acetylcholine levels through the inhibition of hydrolysis of acetylcholine by AChE enzyme. Although showing neuroprotective efficacy in several in vitro and animal models, in vivo studies have reported that it is extensively metabolized upon absorption from the gut, affecting its bioavailability. It also has low BBB penetrability, thus limiting its efficacy in combating neurodegenerative disorders. Therefore, future clinical trials of quercetin and its analogs as neuroprotective agents must improve its bioavailability, developing related molecules with greater gut and brain penetrability, which will likely improve clinical efficacy.

## Figures and Tables

**Figure 1 biomolecules-10-00059-f001:**
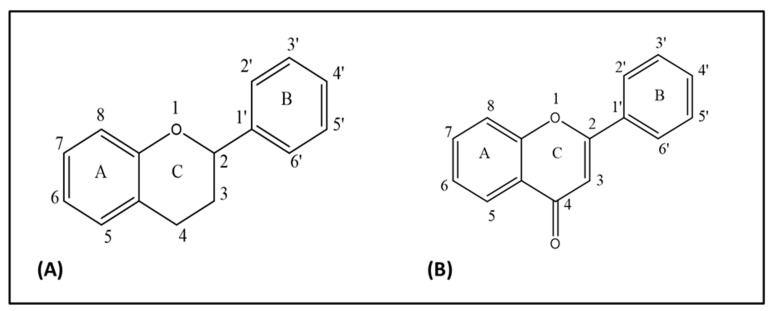
(**A**) Flavan nucleus, (**B**) 4-oxo-flavonoid nucleus.

**Figure 2 biomolecules-10-00059-f002:**
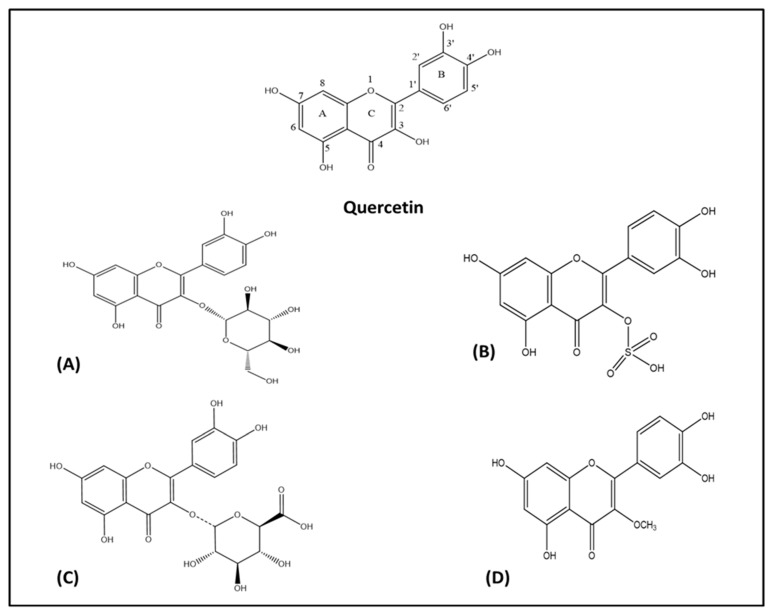
Chemical skeleton of quercetin and derivatives. (**A**) Quercetin glucoside, (**B**) quercetin-3-*O*-sulfate, (**C**) quercetin-3-*O*-glucuronide, and (**D**) 3-*O*-methyl quercetin.

**Figure 3 biomolecules-10-00059-f003:**
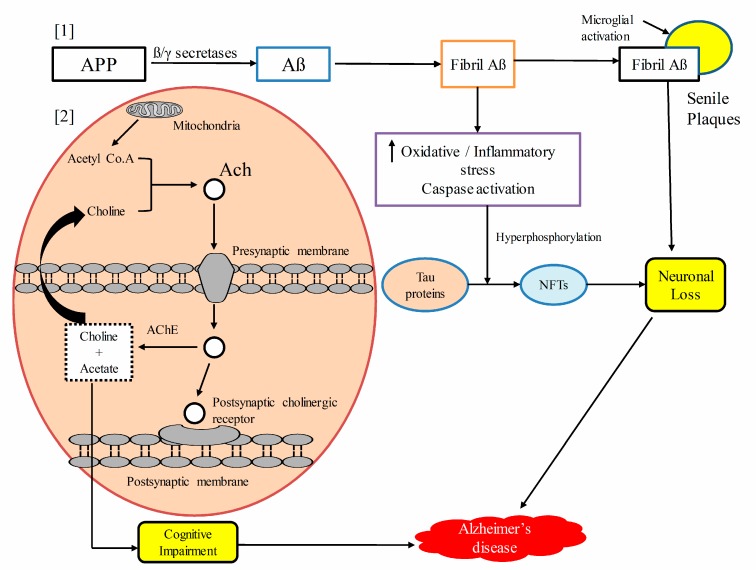
Schematic presentation of the pathogenesis of Alzheimer’s disease. [1] Amyloid precursor protein (APP) is hydrolyzed by β and γ secretases to form β-amyloid (Aβ), which aggregates to form fibril Aβs. Fibril Aβs upregulate oxidative stress, the inflammatory cascade, and caspase activation, which results into the hyperphosphorylation of the Tau protein to form neurofibrillary tangles(NFTs), and the ultimate result is neuronal cells loss. Extensive fibrils along with activated microglia accumulated to form senile plaques, which lead to neuronal and synaptic loss. [2] Upstream regulating acetyl cholinesterase (AChE) enzyme promotes acetylcholine (Ach) degradation, resulting in neurotransmitter deficit, which leads to cognitive impairment. Amyloid precursor protein (APP), amyloid beta proteins (Aβ), neurofibrillary tangles (NFTs), acetylcholine (Ach), acetyl cholinesterase (AChE).

**Figure 4 biomolecules-10-00059-f004:**
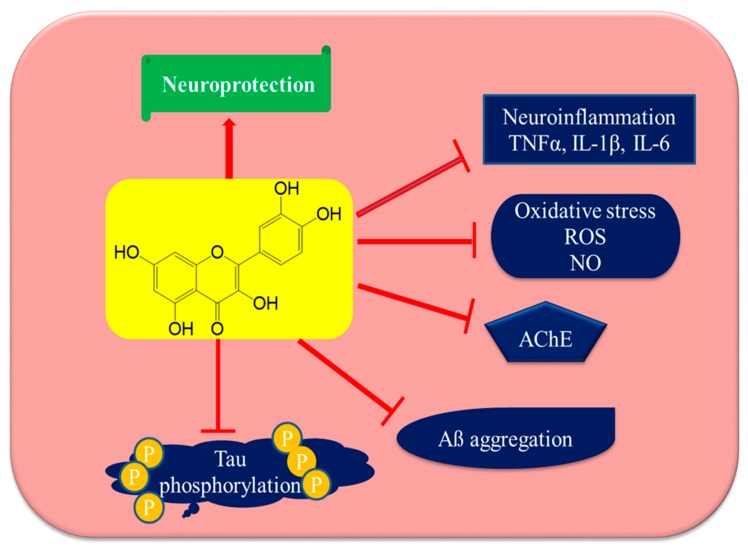
Anti-Alzheimer’s disease targets of quercetin. Quercetin may utilize several mechanistic targets for neuroprotection in Alzheimer’s disease such as the downstream regulation of oxidative stress and neuroinflammation, which leads to the direct protection of neurons, inhibiting AChE enzymes and resulting in increasing acetylcholine levels and reducing Tau phosphorylation and Aβ aggregation. Tumor necrosis factor-α (TNFα), interleukin-1β (IL-1β), interleukin-6 (IL-6), reactive oxygen species (ROS), nitric oxide (NO), acetyl cholinesterase (AChE), and amyloid beta protein (Aβ).

**Figure 5 biomolecules-10-00059-f005:**
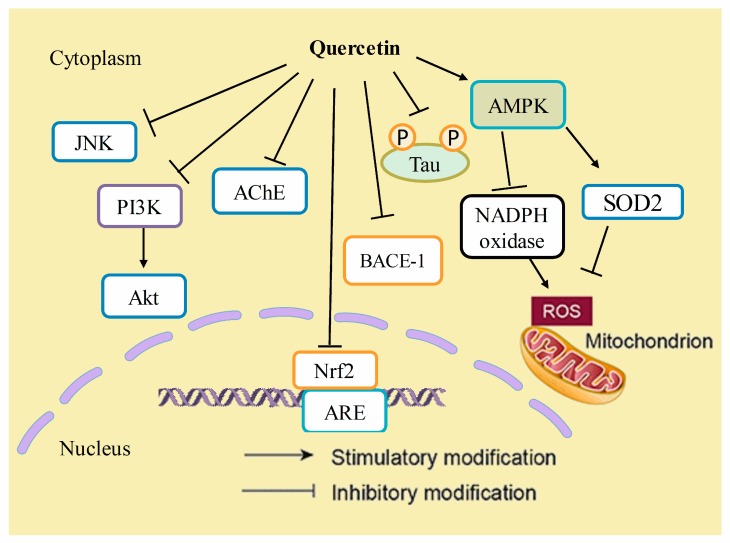
Mechanistic insights of quercetin in Alzheimer’s disease. Quercetin has inhibitory effects on JNK, PI3K/Akt pathways, acetyl cholinesterase (AChE), nuclear factor (erythroid-derived 2)-like 2 (Nrf-2), beta-secretase-1 (BACE-1) enzyme activity, and the hyperphosphorylation of tau proteins. On the other hand, it stimulates the expression of AMP-activated protein kinase (AMPK), which thereby decreases reactive oxygen species (ROS) generation by inhibiting NADPH oxidase activity or by increasing the antioxidant activity of enzymes such as superoxide dismutase-2 (SOD2).

**Table 1 biomolecules-10-00059-t001:** Botanical sources of quercetin.

S.No.	Botanical Name	Family	Common Name	Active Parts	References
01	*Punica granatum*	*Lythraceae*	*Pomegranate*	Fruits	[51]
02	*Ruta graveolens*	*Rutaceae*	*Rue*	Leaves	[51]
03	*Camellia sinensis*	*Theaceae*	*Green tea*	Leaves	[52,53]
04	*Allium cepa*	*Amaryllidaceae*	*Red onion*	Fruits	[54,55]
05	*Mangifera indica*	*Anacardiaceae*	*Mango*	Fruits	[56]
06	*Moringa oleifera*	*Moringaceae*	*Drumstick tree*	Leaves	[57,58]
07	*Cydonia oblonga*	*Rosaceae*	*Quince*	Fruits and leaves	[59]
08	Solidago *canadensis* L.	*Compositae/ Asteraceae*	*Goldenrod*	Flowering parts	[60]
09	*Vaccinium angustifolium* and *Vaccinium corymbosum*	*Ericaceae*	*Blueberries*	Fruits	[61,62]
10	*Phaleria macrocarpa*	*Thymelaceae*	*Mahkotadewa*	Seeds	[63]
11	*Lepidium latifolium*	*Brassicaceae*	*Papperweed*	Roots and leaves	[64,65]
12	*Achras sapota (Manilkara zapota)*	*Sapotaceae*	*Sapodilla*	Fruits	[66]
13	*Cichorium intybus*	*Compositae/ Asteraceae*	*Chicory*	Leaves	[67]
14	*Solanum lycopersicum*	*Solanaceae*	*Tomato*	Fruits	[68]
15	*Malus domestica*	*Rosaceae*	*Apple*	Fruits	[69]
16	*Vitis vinifera*	*Vitaceae*	*Grapevines*	Fruits	[70,71]
17	*Rhamnus alaternus*	*Rhamnaceae*	*Buckthorn*	Bark	[72]
18	*Passiflora incarnate*	*Passifloraceae*	*Passion flower*	Leaves	[73]
19	*Morus alba*	*Moraceae*	*White mulberry or Tut*	Leaves	[74,75]
20	* Ginkgo biloba *	*Ginkgoaceae*	*Maidenhair tree*	Leaves	[76,77]
21	*Hypericum perforatum*	*Hypericaceae*	*St. John’s wort or hypericum*	Aerial parts	[78]
22	Achillea millefolium L.	*Compositae/ Asteraceae*	*Yarrow*	Flowering tops	[79]

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
