# Peer review of "Neuroprotective Effects of Quercetin in Alzheimer’s Disease"

_biomolecules, 2019, doi:10.3390/biom10010059_

Round 1
Reviewer 1 Report
The manuscript of Haroon Khan et al. summarizes the newest results on the neuroprotective effect of quercetin. The authors cite 134 scientific papers int he review. Although the manuscript is a good summary of the topic, there are several mistakes and some important literature data are not mentioned. The main problems are as follows:
The first paragraph (lines 28-40) of the Introduction is too long and contains unnecessary data. The importance of AD is well known for the reader. 2: the legend is not correct. Structure A) shows the quercetin glucoside and not glycoside , the latter is a more general name. The name of D): the site of the methylation ought to be given, as the substance possesses altogether five OH groups. Line 140: as Quercus means oak in English, the end of the sentence could be ”…after Quercus (oak)”. In the paragraph on „Pathophysiological mechanism of AD” at least one modern and well accepted hypothesis of the AD (authors: E. Karran, De Strooper, 2016; the cellular hypothesis on the role of neurons, microglia and astrocytes in progression of AD) ought te be cited. Lines 158-161: the authors should mention, that (at least) two different enzymatic processing of APP occurs in the cells: the amyloidogenic and the non-amyloidogenic pathways. Figure 3. is very simplified: the Abeta and the tau pathways are connected and it ought to be marked. Line 244-246: the hydroxyl group is hydrophilic and can not directly participate in hydrophobic interactions. However, phenolic hydroxyls increase the electron density in the aromatic rings and thus may increase the binding of quercetin with the aromatic amino acids of the peptide beta-sheet structures. In paragraph 7. (in vivo studies): it is not mentioned, which kind of animal (or human) model studies have been used.
There are plenty of typos int he text, some of them are disturbing and should be corrected. Some examples:
Line 47: N-methyl-D-aspartate is the correct name for NMDA. (Aspartame is a synthetic dipeptide, a sweetener).
Line 437-48: memantine is the name of the drug and not meantime.
Line 53: perhaps tacrine and not terrine.
Line 57: correctly Phytochemical
Line 85: heterocyclic pyran (two distinct words)
Line 88: subclasses (one word)
Line 91: flavone class (and not in plural form: flavones).
Line98: quercetin glucoside is correct (glycoside represents a large group of compounds)
Line 120: correctly: C-ring-2,3 double bond.
Line136: „ C4’ improve
Table 1, line 12: the correct name is Cichorium intybus
line 19: „ „ „ is Ginkgo biloba.
Line 211: the correct name is glucuronated
Author Response
Dear Editor/Reviewer
Many thanks for the intensive evaluation of our joint review that leads to several quarries. We religiously address to all those quarries and I am sure their incorporation will greatly ad to the strength of our review.
Reviewer 1:
Comments and Suggestions for Authors
The manuscript of Haroon Khan et al. summarizes the newest results on the neuroprotective effect of quercetin. The authors cite 134 scientific papers in the review. Although the manuscript is a good summary of the topic, there are several mistakes and some important literature data are not mentioned. The main problems are as follows:
The first paragraph (lines 28-40) of the Introduction is too long and contains unnecessary data. The importance of AD is well known for the readerReply: The unnecessary data from first paragraph of introduction have been omitted as per suggestion.
Figure 2: the legend is not correct. Structure A) shows the quercetin glucoside and not glycoside, the latter is a more general name. The name of D): the site of the methylation ought to be given, as the substance possesses altogether five OH groups.
Reply: The terminology “glycoside” has been replaced with glucoside in the legend of Figure 2. The site of the methylation has also been given in (D).
Line 140: as Quercus means oak in English, the end of the sentence could be ”…after Quercus (oak)”.
Reply: The sentence has been restructured as suggested.
In the paragraph on „Pathophysiological mechanism of AD” at least one modern and well accepted hypothesis of the AD (authors: E. Karran, De Strooper, 2016; the cellular hypothesis on the role of neurons, microglia and astrocytes in progression of AD) ought to be cited.
Reply: As suggested, in the paragraph on „Pathophysiological mechanism of AD”, a detailed note on neuroinflammation and the role of neurons, microglia and astrocytes in progression of AD has been added and highlighted.
Lines 158-161: the authors should mention that (at least) two different enzymatic processing of APP occurs in the cells: the amyloidogenic and the non-amyloidogenic pathways.
Reply: The two different enzymatic processing of APP (the amyloidogenic and the non-amyloidogenic pathways) occurs in the cell have been mentioned as suggested.
Figure 3 is very simplified: the Abeta and the tau pathways are connected and it ought to be marked.
Reply: As suggested, Figure 3 has been revised.
Line 244-246: the hydroxyl group is hydrophilic and cannot directly participate in hydrophobic interactions. However, phenolic hydroxyls increase the electron density in the aromatic rings and thus may increase the binding of quercetin with the aromatic amino acids of the peptide beta-sheet structures.
Reply: Lines 244-246 have been re-structured as suggested.
In paragraph 7. (In vivo studies): it is not mentioned, which kind of animal (or human) model studies have been used?Reply: In paragraph 7 (In vivo studies), the kind of study model (animal or human) have mentioned as suggested.
There are plenty of typos in the text, some of them are disturbing and should be corrected. Some examples: Line 47: N-methyl-D-aspartate is the correct name for NMDA. Line 437-48: memantine is the name of the drug and not meantime. Line 53: perhaps tacrine and not terrine. Line 57: correctly Phytochemical Line 85: heterocyclic pyran (two distinct words) Line 88: subclasses (one word) Line 91: flavone class (and not in plural form: flavones). Line98: quercetin glucoside is correct (glycoside represents a large group of compounds) Line 120: correctly: C-ring-2,3 double bond. Line136: „ C4’ improve Table 1, line 12: the correct name is Cichorium intybus Line 19: „ „ „ is Ginkgo biloba. Line 211: the correct name is glucuronated
Reply: The typo mistakes in the text have been corrected as suggested and highlighted.
Reviewer 2 Report
In this manuscript, Haroon Khan and colleagues have reviewed the current literature related to the neuroprotective effects of quercetin in Alzheimer’s disease. The article covers several topics, including the chemistry of this flavonoid, the natural sources, the pathophysiology of Alzheimer’s disease, and the mechanisms of action of this compound. This is an interesting topic. Nevertheless, it has the main limitation that there some review articles that have already been published on this topic and part of the information that is contained in this article partially overlaps some of these reviews (ex. reference 27, 134). It was surprising to find that a lot of the references used in this manuscript were reviews and, contrarily to what it is usually seen in review articles, that there was a lack of a systematic summary of research articles (ex. performed in cells, animals or humans) evaluating the neuroprotective effects of quercetin (typically summarized in the form of a Table) to provide evidence about its neuroprotective activities. Additionally, I had the feeling that there was a contrast (and a slight lack of connection) between two parts of the review: the first part of the review (subheadings 1-5) was very optimistic on the neuroprotective effects of quercetin, whereas the last part (subheadings 6, 7 and conclusions) had a more realistic and balanced view (explaining both the promising vision and also the current limitations evidenced by the literature).
Major concerns:
In several parts of the manuscript, the information from basic studies (ex. performed in vitro) is not clearly differentiated from the information from preclinical studies (experiments performed in animal models), from information performed in clinical trials. Given the fact that these pieces of information have different degrees of scientific evidence, this information should be clearly differentiated, explained or at least mentioned. There are several examples in which this occurs. For instance, in lines 131-137, when the activity of methylated metabolites of quercetin is explained, it is stated that specific substitutions result in improved neuroprotective activities. How this information was obtained? Which is the degree of scientific evidence? The information from in vitro/ex vivo experiments is not the same that the information that stems from in vivo studies and this should be clearly differentiated. In subheadings 6 (in vitro) and 7 (in vivo) this differentiation is performed, but the previous subheadings contain a mixture of information in which the origin of the study (in vitro vs in vivo) is not always clearly described with enough detail. There is relevant information that is missing on the topic related to the bioavailability and pharmacokinetics of quercetin and their metabolites. While the authors mention that quercetin undergoes extensive metabolism (methylation, sulfation, and glucuronidation) but they do not describe with enough detail the process of absorption (Tmax, plasma concentrations…), distribution (especially to the central nervous system), metabolism and excretion of quercetin. This lacking information is required because published studies have shown that quercetin cannot be detected in plasma or in the brain after oral intake and this is an important limitation when explaining the neuroprotective effects of this compound. Also related to the previous comment, as explained by the authors, when given at high doses, quercetin has pro-oxidant activities, meaning that the knowledge on the doses and/or plasma circulating concentrations are important. However, the authors do not mention which are the plasma concentrations found after the intake of quercetin. There is also missing information related to the stability of this compound. The presence of multiple hydroxyl groups, which make this compound a very good antioxidant, may also compromise its chemical stability. Information on this topic was missing. Figure 5 contains information about specific mechanisms that are not present in the text (ex. Uncoupling protein 2 or UCP2) As explained before, there was a lack of connection between the first part of the manuscript (subheadings 1-5) and the second one (subheadings 6-7). Some of the information was repeated and/or explained in a different way. Lines 47 and 48: “meantime” is not the correct name of the drug. The authors refer to memantine. Line 53: “terrine” is not the correct name of the drug. The authors refer to tacrine.
Minor concerns
Line 29: 65 years of age or older Line 60: flavonoids are further divided in 6 classes. Which ones? Line 56: This article does not corroborate the ability of flavonoids to traverse the BBB. It says that it has been suggested that flavonoids cross the BBB. Although some in vitro and preclinical studies, support this idea, the article also states that "different flavonoid subclasses differ in their ability to cross the BBB" Line 68: references that Support this idea should be added Lines 69-70: reference 26 does not show that quercetin is the most potent anti-oxidant of plant origin, it says that "quercitin is one of the most potent antioxidants of plant origin" Line 85: heterocyclic pyran Line 85: Please revise the sentence Lines 131-137: As explained before, these studies were in vitro or ex vivo. It should be differentiated a biological activity that takes place in vitro, from that one that has been also confirmed in preclinical studies, from that that has been confirmed in humans. The degree of scientific evidence is different and should be taken into account. Line 143: The authors say that “Plant families rich in quercetin are Compositae” but none of the 20 species that appear in Table 1 belong to this family. Please add some representative examples. Line 148: This subheading does not have a number. Line 181: Given the fact that this article is focused on the neuroprotective role of quercetin on Alzheimer’s, the term MCI should be described in more detail. Line 219: The authors have not explained before that isoquercetin it the 3-O-glucoside of quercetin. Line 282: Please revise the sentence as, it has been explained before in this same manuscript that tacrine has problems of hepatotoxicity. Moreover, in reference 113 it does not say that tacrine is presently undergoing trials to evaluate the antiamnesic activity. Line 300: “quercetinalso” should appear as “quercetin also” Line 331: “microphages”, the authors probably refer to macrophages Line 385: It is good to explain that quercetin has a low bioavailability but this information would have been useful if it had been described before. Line 391: This review (reference 134) discusses the most recent data on the potential of quercetin to confer neuroprotection. Unfortunately, most of the in vitro studies have used quercetin aglycone, which is not detectable in the plasma or in the brain after oral intake. Moreover, quercetin metabolites and glycosides seem to be less neuroprotective and penetrate the BBB less efficiently than aglycone. The format of the references should be carefully reviewed: ex. reference 1, reference 7 (the author is not correct), reference 18, 27, 31, 32, 44, 45, 48, 53… (check pages).
Author Response
Dear Editor/Reviewer
Many thanks for the intensive evaluation of our joint review that leads to several quarries. We religiously address to all those quarries and I am sure their incorporation will greatly ad to the strength of our review.
Reviewer 2:
Comments and Suggestions for Authors
In this manuscript, Haroon Khan and colleagues have reviewed the current literature related to the neuroprotective effects of quercetin in Alzheimer’s disease. The article covers several topics, including the chemistry of this flavonoid, the natural sources, the pathophysiology of Alzheimer’s disease, and the mechanisms of action of this compound. This is an interesting topic.
1. Nevertheless, it has the main limitation that there some review articles that have already been published on this topic and part of the information that is contained in this article -partially overlaps some of these reviews (ex. reference 27, 134).
Reply: As suggested, this could be a limitation that current Review on neuroptective effects of Quercetin does overlap with previous published Review articles on same topic. It is likely to be mention that previous published Review articles focuses on one specific theme such as in Reference 27, authors were focusing on attenuation of oxidative stress by quercetin as possible mechanism of neuroprotection and in Reference 134 was focusing on pharmacokinetics studies of quercetin, In vitro and in vivo experimental studies of quercetin in relation to Neuroprotection. In contrast, in present review authors give a detail picture of neuroprotective potential of quercetin along with limitations to its clinical use for same purpose. Beside countering oxidative stress, authors are exploring comprehensively the neuroprotective mechanisms of quercetin in Alzheimer’s disease.
2. It was surprising to find that a lot of the references used in this manuscript were reviews and, contrarily to what it is usually seen in review articles, that there was a lack of a systematic summary of research articles (ex. performed in cells, animals or humans) evaluating the neuroprotective effects of quercetin (typically summarized in the form of a Table) to provide evidence about its neuroprotective activities.
Reply: As suggested, original researches have been cited where possible especially in sections “Anti-Alzheimer’s disease mechanisms of quercetin, In vitro studies on the efficacy of quercetin and In vivo studies on the efficacy of quercetin”.
3. Additionally, I had the feeling that there was a contrast (and a slight lack of connection) between two parts of the review: the first part of the review (subheadings 1-5) was very optimistic on the neuroprotective effects of quercetin, whereas the last part (subheadings 6, 7 and conclusions) had a more realistic and balanced view (explaining both the promising vision and also the current limitations evidenced by the literature).
Reply: As suggested, we tried to fill a gap between the connectivity of two parts, as Part 1 (subheadings 1-5) was previously optimistic, we currently also addresses the limitations of quercetin in its clinical use under subheading “Pharmacokinetics of Quercetin” (highlighted).
Major concerns:
1. In several parts of the manuscript, the information- from basic studies (ex. performed in vitro) is not clearly differentiated from the information from preclinical studies (experiments performed in animal models), from information performed in clinical trials. Given the fact that these pieces of information have different degrees of scientific evidence, this information should be clearly differentiated, explained or at least mentioned. There are several examples in which this occurs. For instance, in lines 131-137, when the activity of methylated metabolites of quercetin is explained, it is stated that specific substitutions result in improved neuroprotective activities. How this information was obtained? Which is the degree of scientific evidence? The information from in vitro/ex vivo experiments is not the same that the information that stems from in vivo studies and this should be clearly differentiated. In subheadings 6 (in vitro) and 7 (in vivo) this differentiation is performed, but the previous subheadings contain a mixture of information in which the origin of the study (in vitro vs in vivo) is not always clearly described with enough detail.
Reply: As suggested, in vitro and in vivo studies have been mentioned in the manuscript which makes it simple and easier to distinguish between in vitro, in vivo and human studies (hughlighted).
2. There is relevant information that is missing on the topic related to the bioavailability and pharmacokinetics of quercetin and their metabolites. While the authors mention that quercetin undergoes extensive metabolism (methylation, sulfation, and glucuronidation) but they do not describe with enough detail the process of absorption (Tmax, plasma concentrations…), distribution (especially to the central nervous system), metabolism and excretion of quercetin. This lacking information is required because published studies have shown that quercetin cannot be detected in plasma or in the brain after oral intake and this is an important limitation when explaining the neuroprotective effects of this compound.
Reply: The pharmacokinetic parameters in relation to absorption, Tmax, Cmax, metabolism, clearance and distribution (especially to the central nervous system) have been described under separate section “Pharmacokinetic parameters of quercetin”.
3. Also related to the previous comment, as explained by the authors, when given at high doses, quercetin has pro-oxidant activities, meaning that the knowledge on the doses and/or plasma circulating concentrations are important. However, the authors do not mention which are the plasma concentrations found after the intake of quercetin.
Reply: The low and high doses of quercetin used previously in In vitro study have been added to the paragraph which relates ani-oxidant and toxic effects of quercetin with its dose (highlighted).
4. There is also missing information related to the stability of this compound. The presence of multiple hydroxyl groups, which make this compound a very good antioxidant, may also compromise its chemical stability. Information on this topic was missing.
Reply: The impact of multiple hydroxyl groups on photo-stability of quercetin have been discussed under section “Chemistry of Quercetin” (highlighted).
5. Figure 5 contains information about specific mechanisms that are not present in the text (ex. Uncoupling protein 2 or UCP2).
Reply: Figure 5 has been revised and the information not present in the Manuscript have been eliminated.
6. As explained before, there was a lack of connection between the first part of the manuscript (subheadings 1-5) and the second one (subheadings 6-7). Some of the information was repeated and/or explained in a different way.
Reply: Needful changes have made in the revised version.
7. Lines 47 and 48: “meantime” is not the correct name of the drug. The authors refer to memantine. Line 53: “terrine” is not the correct name of the drug. The authors refer to tacrine.
Reply: There were number of typo mistakes in the manuscript and typo mistakes have been corrected as suggested.
Minor concerns
1. Line 29: 65 years of age or older
Reply: Sentence have been corrected and restructured as suggested.
2. Line 60: flavonoids are further divided in 6 classes. Which ones?
Reply: The names of 6 classes of flavonoid have been mentioned as suggested.
3. Line 65: This article does not corroborate the ability of flavonoids to traverse the BBB. It says that it has been suggested that flavonoids cross the BBB. Although some in vitro and preclinical studies support this idea, the article also states that "different flavonoid subclasses differ in their ability to cross the BBB".
Reply: As suggested, the sentences have been revised and highlighted.
4. Line 68: references that Support this idea should be added
Reply: As suggested, References have been added to support the idea explained in line 68.
5. Lines 69-70: reference 26 does not show that quercetin is the most potent anti-oxidant of plant origin, it says that "quercitin is one of the most potent antioxidants of plant origin" .
Reply: As suggested the sentence has been corrected and restructured.
6. Line 85: heterocyclic pyran
Reply: It was a typo mistake and has been corrected and highlighted.
7. Line 85: Please revise the sentence
Reply: Sentence has been revised and highlighted.
8. Lines 131-137: As explained before, these studies were in vitro or ex vivo. It should be differentiated a biological activity that takes place in vitro, from that one that has been also confirmed in preclinical studies, from that that has been confirmed in humans. The degree of scientific evidence is different and should be taken into account.
Reply: In vitro / in vivo studies have been mentioned o the studies as suggested.
9. Line 143: The authors say that “Plant families rich in quercetin are Compositae” but none of the 20 species that appear in Table 1 belong to this family. Please add some representative examples.
Reply: Some examples of “Compositae Plant families rich in quercetin” have been added to the species appear in Table 1, as suggested.
10. Line 148: This subheading does not have a number.
Reply: Number have been added to subheading “Pathophysiological mechanism of Alzheimer’s disease” and highlighted.
11. Line 181: Given the fact that this article is focused on the neuroprotective role of quercetin on Alzheimer’s, the term MCI should be described in more detail.
Reply: As suggested, the world MCI has been described in detail in the same paragraph and highlighted.
12. Line 219: The authors have not explained before that isoquercetin it the 3-O-glucoside of quercetin.
Reply: As suggested, the glycosidal forms of quercetin including isoquercetin (3-O-glucoside) has been mentioned/explained under section 3 [Sources] and highlighted.
13. Line 282: Please revise the sentence as, it has been explained before in this same manuscript that tacrine has problems of hepatotoxicity. Moreover, in reference 113 it does not say that tacrine is presently undergoing trials to evaluate the antiamnesic activity.
Reply: The sentence has been revised and corrected as suggested and highlighted.
14. Line 300: “quercetin also” should appear as “quercetin”
Reply: It was a typo mistake and has been corrected and highlighted.
15. Line 331: “microphages”, the authors probably refer to macrophages
Reply: It was a typo mistake and has been corrected and highlighted.
16. Line 385: It is good to explain that quercetin has a low bioavailability but this information would have been useful if it had been described before.
Reply: As suggested, information regarding low bioavailability of quercetin has been briefly explained under section “Pharmacokinetic parameters of quercetin” and highlighted.
17. Line 391: This review (reference 134) discusses the most recent data on the potential of quercetin to confer neuroprotection. Unfortunately, most of the in vitro studies have used quercetin aglycone, which is not detectable in the plasma or in the brain after oral intake. Moreover, quercetin metabolites and glycosides seem to be less neuroprotective and penetrate the BBB less efficiently than aglycone.
Reply: As suggested, the distribution of quercetin aglycone and metabolites has been discussed in detail under section “Pharmacokinetic parameters of quercetin”.
18. The format of the references should be carefully reviewed: ex. reference 1, reference 7 (the author is not correct), reference 18, 27, 31, 32, 44, 45, 48, 53… (check pages).
Reply: As suggested, References have been revised.
Round 2
Reviewer 2 Report
The authors have performed major revisions according to the previous comments and the quality of the review has significantly improved. Now the tone is much more balanced and it provides a more critical and unbiased review of the neuroprotective effects of quercitin in Alzheimer's disease. Remaining aspects that I would recommend improving before publication are:
1) Figure 2 contains important mistakes (2B is not quercitin sulfate; 2C is not quercitin glucuronide)
2) I would recommend improving the quality of Figure 3
3) The text of Figure 5 cannot be read
Author Response
Dear Editor
Many thanks for the important suggestion. We duly addressed all of them.
Comments and Suggestions for Authors
The authors have performed major revisions according to the previous comments and the quality of the review has significantly improved. Now the tone is much more balanced and it provides a more critical and unbiased review of the neuroprotective effects of quercitin in Alzheimer's disease. Remaining aspects that I would recommend improving before publication are:
Figure 2 contains important mistakes (2B is not quercitin sulfate; 2C is not quercitin glucuronide)Reply: As suggested, mistakes in Figure 2 have been corrected and highlighted in the caption of the figure.
I would recommend improving the quality of Figure 3
Reply: As suggested, the quality of Figure 3 has been improved.
The text of Figure 5 cannot be read
Reply: As suggested, the boxes in Figure 5 have been reshaped and readjusted, which makes the text of the figure clear.
Prof. Dr. Haroon Khan